# Thermodynamics and Stability of Non-Equilibrium Steady States in Open Systems

**DOI:** 10.3390/e21070704

**Published:** 2019-07-18

**Authors:** Miroslav Bulíček, Josef Málek, Vít Průša

**Affiliations:** Faculty of Mathematics and Physics, Charles University, Sokolovská 83, 18675 Prague, Czech Republic

**Keywords:** nonlinear stability, Lyapunov functional, thermodynamics, thermodynamically open systems, 35Q79, 37L15, 37B25

## Abstract

Thermodynamical arguments are known to be useful in the construction of physically motivated Lyapunov functionals for nonlinear stability analysis of spatially homogeneous equilibrium states in thermodynamically isolated systems. Unfortunately, the limitation to isolated systems is essential, and standard arguments are not applicable even for some very simple thermodynamically open systems. On the other hand, the nonlinear stability of thermodynamically open systems is usually investigated using the so-called energy method. The mathematical quantity that is referred to as the “energy” is, however, in most cases not linked to the energy in the physical sense of the word. Consequently, it would seem that genuine thermo-dynamical concepts are of no use in the nonlinear stability analysis of thermodynamically open systems. We show that this is not the case. In particular, we propose a construction that in the case of a simple heat conduction problem leads to a physically well-motivated Lyapunov type functional, which effectively replaces the artificial Lyapunov functional used in the standard energy method. The proposed construction seems to be general enough to be applied in complex thermomechanical settings.

## 1. Introduction

### 1.1. Stability of Spatially Homogeneous Equilibrium States in Thermodynamically Isolated Systems

Classical thermodynamics of continuous media can be gainfully exploited in nonlinear stability analysis of thermodynamically isolated systems. The physical concepts of net total energy $E_{\mathrm{tot}}$ and the net entropy *S* help one to design natural Lyapunov functionals for nonlinear stability analysis, provided that one is interested in *spatially homogeneous equilibrium rest states* in *thermodynamically isolated* systems. For example, if one is interested in the stability of equilibrium rest state of an incompressible Navier–Stokes fluid in a thermodynamically isolated vessel, then one can introduce the functional
(1)$$S-\frac{1}{\theta_{\mathrm{bdr}}}E_{\mathrm{tot}},$$
where $\theta_{\mathrm{bdr}}$ denotes the temperature value at the equilibrium, and this functional turns out to be a natural Lyapunov functional characterising the stability of the rest state.

Functionals of the type (Equation 1) are useful even for very simple *thermodynamically open* systems, namely for systems where the temperature value $\theta_{\mathrm{bdr}}$ on the boundary is spatially homogeneous, see the seminal contribution by Coleman [1] and the comprehensive treatise by Gurtin [2]. However, if the temperature value $\theta_{\mathrm{bdr}}$ on system boundary varies in space, then the standard construction of Lyapunov functional as introduced by Coleman [1] is inapplicable. Indeed, if $\theta_{\mathrm{bdr}}$ is a function of position, then expressions of the type (Equation 1) do not even define a functional.

This restriction is very limiting, since it prevents one from using the construction in the stability analysis of very simple *thermodynamically open* systems such as heat conduction in a differentially heated rigid body. Consequently, nonlinear stability analysis of spatially inhomogenous non-equilibrium states in thermodynamically open systems is beyond the reach of the methods based on the functional (Equation 1).

### 1.2. Stability of Spatially Inhomogeneous Non-Equilibrium States in Thermodynamically Open Systems; Energy Method and Its Deficiencies

A mathematical method referred to as the *energy method* has been developed in order to deal with the nonlinear stability of *inhomogenous non-equilibrium states in thermodynamically open systems*. The method has been used in numerous works on stability of solutions to systems of nonlinear partial differential equations, see for example Joseph [3], Joseph [4] and Straughan [5] and references therein, and it became the standard method in the field.

The method originated in hydrodynamic stability problems, see Reynolds [6] and Orr [7], and it was popularised and further elaborated by Serrin [8]. In the original hydrodynamic stability setting the link between the mathematical technique and its physical underpinning is very clear. In hydrodynamic stability problems, the quantity of interest in the mathematical stability analysis is the square of the Lebesgue norm ${∥v∥}_{L^{2}\left(\Omega \right)}$ of the velocity field v. This quantity is tantamount, up to a constant, to the kinetic energy of the fluid occupying the domain of interest Ω. Consequently, the name energy method is well justified, and one can happily contemplate the close interplay between physics and mathematics.

However, if the *energy method* is used in a more complex setting such as the nonlinear stability of thermal convection, the name energy method becomes problematic. For example, Straughan [5] in his discussion on stability of thermal convection states that:
We consider the simplest, natural “energy”, formed by adding the kinetic and thermal energies of perturbations, and so define $E\left(t\right)=\frac{1}{2}{∥u∥}_{}^{2}+\frac{1}{2}\mathrm{Pr}{∥\theta ∥}_{}^{2}$.

(Please note the quotation marks.) Similarly, Joseph [4] in his discussion of nonlinear stability of thermosolutal convection states that:
Though $\frac{\langle {\left|u\right|}^{2}\rangle }{2}$ is proportional to the kinetic energy, the other quadratic integrals $\langle \theta^{2}\rangle$ and $\langle \gamma^{2}\rangle$ cannot be called energies in any strict sense.

These authors restrain themselves from unequivocally using the word energy for a very good reason. The volume integrals of the square of temperature field, that is the integrals ${∥\theta ∥}_{}^{2}{=}_{\mathrm{def}}\int_{\Omega }\theta^{2}\;dv$ and $\langle \theta^{2}\rangle {=}_{\mathrm{def}}\frac{1}{\left|\Omega \right|}\int_{\Omega }\theta^{2}\;dv$, have no clear physical interpretation. In particular, they do not have the meaning of *thermal energy*. This is in striking contrast with the volume integrals ${∥u∥}_{}^{2}{=}_{\mathrm{def}}\int_{\Omega }{\left|u\right|}^{2}\;dv$ and $\langle u^{2}\rangle {=}_{\mathrm{def}}\frac{1}{\left|\Omega \right|}\int_{\Omega }{\left|u\right|}^{2}\;dv$ of the velocity field u that are, up to a constant, identical to the *kinetic energy*.

This shows that the name *energy method* is in these settings inappropriate and misleading, although the mathematical results obtained on the basis of the energy method are of course valid. The problem is that the quantity referred to as the energy is no longer the energy in the physical sense of the word, and it seems to be a quantity designed artificially on the basis of mathematical convenience. The former *clear link between the mathematical method and physics is lost*.

### 1.3. Stability of Spatially Inhomogeneous Non-Equilibrium States in Thermodynamically Open Systems—A Search for Novel Construction of a Physically Motivated Lyapunov Type Functional

The question is whether the presence of the volume integrals of the square of the temperature field ${∥\theta ∥}_{L^{2}\left(\Omega \right)}^{2}$ and $\langle \theta^{2}\rangle$, can be explained/justified by appealing to some other physical concepts than the energy. If this is not possible, one can ask whether there exists another functional of the temperature field that is physically motivated and that can be effectively used as a Lyapunov functional. Further, using such a functional one should be at least able to reach the same conclusions concerning the stability problem for the non-equilibrium steady state in the given *thermodynamically open system* as the conclusions that can be obtained by the standard energy method.

Ideally, the construction of a suitable Lyapunov functional for thermodynamically *open* systems should be from the physical point of view as transparent as in the case of thermodynamically *closed* systems, see Coleman [1]. One would like to again see a clear connection between physics and the corresponding mathematical method.

Since the core problem is the use of the standard energy method in heat conduction problems, we investigate the questions in a simple setting of heat conduction in a rigid body. We propose a procedure that leads to the construction of a *physically well motivated Lyapunov type functional that regarding the nonlinear stability analysis of a non-equilibrium steady state effectively replaces the artificial squared Lebesgue norm of temperature field*.

Using the newly designed Lyapunov type functional, we recover the standard stability result for the heat conduction problem in a rigid body. Heat conduction governed by the standard Fourier law. This is of course not a fundamental result. The main outcome of the presented analysis is different.

The physically motivated Lyapunov type functional for a non-equilibrium steady state in a thermodynamically *open* system is systematically constructed using the physically motivated Lyapunov type functional for the equilibrium rest state in the corresponding thermodynamically *isolated* system. Consequently, the construction can be seen as a proper generalisation of the standard thermodynamical procedure introduced by Coleman [1]. More importantly, the proposed construction of a physically motivated Lyapunov type functional seems to be *general enough to be applied in more complex thermomechanical settings* than heat conduction.

## 2. Outline

The paper is organised as follows. In Section 3, we introduce the stability problem for the steady solution $\hat{\theta }$ of the heat conduction equation in a rigid body. (Heat conduction governed by the standard Fourier law.) In Section 4 we recall the standard nonlinear stability analysis based on the *energy method*, and we comment in detail on the abuse of the word *energy* in this setting. At the end of Section 4 we rephrase the nonlinear stability analysis as a problem of the design of a suitable Lyapunov functional, which is in the case of the standard energy method given by the formula
(2)$$V_{\mathrm{std}}{=}_{\mathrm{def}}\int_{\Omega }\rho c_{V}{\left(\tilde{\theta }\right)}^{2}\;dv,$$
where $\tilde{\theta }$ denotes the temperature perturbation, see Section 3 for a detailed discussion of the notation.

In Section 5 we propose a physically well motivated construction of a Lyapunov type functional suitable for nonlinear stability analysis. (The functional will be different from the functional $V_{\mathrm{std}}$ used in the *energy method*.) In Section 5.1, we recall basic facts from continuum thermodynamics, and then we use thermodynamical concepts in the nonlinear stability analysis.

First, see Section 5.2, we focus on the stability of the equilibrium rest state in a *thermodynamically isolated* system. (Heat conduction with zero Neumann boundary condition.) The outlined analysis provides an answer to the question *why* one should consider functional (Equation 1) as a natural candidate for a Lyapunov functional. In this sense, it is complementary to the analysis by Coleman [1], who took the functional of type (Equation 1) as given, and then showed that it actually *is* a Lyapunov functional.

Second, see Section 5.3, we focus on the stability of a *non-equilibrium steady state* in a *thermodynamically open* system. (Heat conduction with a spatially inhomogeneous Dirichlet boundary condition.) We use the previously designed Lyapunov type functional for the *thermodynamically isolated* system, and we show how to use this functional in designing a new Lyapunov type functional
(3)$$V_{\mathrm{neq}}=\int_{\Omega }\rho c_{V}\hat{\theta }\left[\frac{\tilde{\theta }}{\hat{\theta }}-\ln\left(1+\frac{\tilde{\theta }}{\hat{\theta }}\right)\right]\;dv,$$
that is suitable for this *thermodynamically open* system. The functional $V_{\mathrm{neq}}$ is argued to be a physically well-justified counterpart of $V_{\mathrm{std}}$. The results obtained are discussed in Section 6.

Finally, see Appendix A, we document the power of the advocated method in the nonlinear stability analysis of heat conduction in a rigid body whose thermal conductivity is a function of temperature. (Heat conduction governed by a *nonlinear* variant of Fourier law.) In this case we are again dealing with a thermodynamically open system, but its dynamics is now governed by a *nonlinear* partial differential equation.

## 3. Stability of Heat Conduction in a Rigid Body

### 3.1. Governing Equation

Let us consider a simple problem of heat conduction in a rigid body that occupies a domain Ω. The evolution of the temperature field θ is governed by the standard heat conduction equation
(4)$$\rho c_{V}\frac{\partial \theta }{\partial t}=\mathrm{div}\left(\kappa \nabla \theta \right),$$
where $c_{V}$ denotes the specific heat capacity at constant volume, $\left[c_{V}\right]=\frac{L^{2}}{T^{2}\cdot Q}$, κ denotes the thermal conductivity, $\left[\kappa \right]=\frac{M\cdot L}{T^{3}\cdot Q}$, and ρ denotes the density, $\left[\rho \right]=\frac{M}{L^{3}}$. All material parameters are assumed to be constant and positive. Once the initial and boundary conditions are specified, one can solve the equation, and obtain the solution hereafter denoted as $\hat{\theta }$. The question is whether the solution $\hat{\theta }$ is stable with respect to perturbations.

The most studied case is the stability of the steady solution $\hat{\theta }$ to (Equation 4) with a prescribed time-independent temperature value $\theta_{\mathrm{bdr}}$ on the boundary. This means that $\hat{\theta }$ solves the problem
(5a)$$\begin{array}{c} 0=\mathrm{div}(\kappa \nabla \hat{\theta }), \end{array}$$
(5b)$$\begin{array}{c} {\left.\hat{\theta }\right|}_{\partial \Omega }=\theta_{\mathrm{bdr}}. \end{array}$$
The solution $\hat{\theta }$ is usually called the *non-equilibrium steady state*, see Section 5.3.1 for the rationale of this nomenclature.

### 3.2. Stability of Steady Solution to the Governing Equation

The *nonlinear stability* of the steady non-equilibrium solution $\hat{\theta }$ essentially means that any time-dependent temperature field of the form $\theta =\hat{\theta }+\tilde{\theta }$ eventually tends to the steady non-equilibrium solution $\hat{\theta }$ as time goes to infinity. In other words, if the temperature field
(6)$$\theta {=}_{\mathrm{def}}\hat{\theta }+\tilde{\theta }$$
solves the initial-boundary value problem
(7a)$$\begin{array}{c} \rho c_{V}\frac{\partial \theta }{\partial t}=\mathrm{div}\left(\kappa \nabla \theta \right), \end{array}$$
(7b)$$\begin{array}{c} {\left.\theta \right|}_{\partial \Omega }=\theta_{\mathrm{bdr}}, \end{array}$$
(7c)$$\begin{array}{c} {\left.\theta \right|}_{t=0}=\theta_{\mathrm{init}}, \end{array}$$
with an initial temperature distribution $\theta_{\mathrm{init}}$, then one says that the steady non-equilibrium solution $\hat{\theta }$ is unconditionally asymptotically stable provided that $\tilde{\theta }\to 0$ as t→+∞ irrespective of the choice of the initial condition. The convergence $\tilde{\theta }\to 0$ is typically understood as the convergence in a Lebesgue space norm, which under the assumptions granting the regularity of the solution, implies also the pointwise convergence everywhere in the domain Ω.

The adjective nonlinear means that we are interested in the stability with respect to finite perturbations, and that we are not dealing with the dynamics of the linearised equations in the neighborhood of the steady state as in the standard linearised stability theory, see for example Lin [9], Chandrasekhar [10], Yudovich [11], Drazin and Reid [12] or Schmid and Henningson [13].

## 4. Unconditional Asymptotic Stability of Steady Non-Equilibrium Solution—The Standard Proof

### 4.1. Standard Energy Method

The standard *energy method* based proof of unconditionally asymptotic stability of a steady non-equilibrium solution $\hat{\theta }$ to (Equation 4) with boundary condition (5b) proceeds as follows.

First, one formulates the governing equations for the perturbation $\tilde{\theta }$. Since $\theta =\hat{\theta }+\tilde{\theta }$ solves (7) and $\hat{\theta }$ solves (5), the governing equations for the perturbation $\tilde{\theta }$ read
(8a)$$\begin{array}{c} \rho c_{V}\frac{\partial \tilde{\theta }}{\partial t}=\mathrm{div}(\kappa \nabla \tilde{\theta }), \end{array}$$
(8b)$$\begin{array}{c} {\left.\tilde{\theta }\right|}_{\partial \Omega }=0, \end{array}$$
(8c)$$\begin{array}{c} {\left.\tilde{\theta }\right|}_{t=0}=\theta_{\mathrm{init}}-\hat{\theta }. \end{array}$$


Second, one multiplies the evolution Equation (8a) by $\tilde{\theta }$, integrates over the domain Ω, and then uses integration by parts in the term $\int_{\Omega }\mathrm{div}(\kappa \nabla \tilde{\theta })\tilde{\theta }\;dv$. The boundary term in the integration by parts formula vanishes in virtue of the boundary condition ([Disp-formula FD8b-entropy-21-00704]), and one obtains the equality
(9)$$\frac{1}{2}\frac{d}{dt}\int_{\Omega }\rho c_{V}{\left(\tilde{\theta }\right)}^{2}\;dv=-\int_{\Omega }\kappa \nabla \tilde{\theta }•\nabla \tilde{\theta }\;dv.$$
Symbol a•b denotes the standard scalar product of two vectors in $R^{3}$. This equation is the evolution equation for the quantity
(10)$$E_{\mathrm{std}}{=}_{\mathrm{def}}\frac{1}{2}\int_{\Omega }\rho c_{V}{\left(\tilde{\theta }\right)}^{2}\;dv,$$
which is commonly referred to as the *energy* of the perturbation $\tilde{\theta }$ or the *energy norm* of the perturbation $\tilde{\theta }$, see for example Joseph [4] or Straughan [5]. Equation (Equation 9) shows that the energy $E_{\mathrm{std}}$ of the perturbation decays in time, $\frac{dE_{\mathrm{std}}}{dt}\leq 0$, which essentially finishes the proof of unconditional asymptotic stability of the solution $\hat{\theta }$.

Moreover, using the standard Poincaré inequality, see for example Gilbarg and Trudinger [14] or Evans [15], it is easy to show that the norm of the perturbation decays to zero in an exponentially fast fashion. A similar argument can be carried out if the Dirichlet boundary condition ([Disp-formula FD5b-entropy-21-00704]) is replaced by the zero Neumann boundary condition ${\left.\nabla \theta •n\right|}_{\partial \Omega }=0$, where n denotes the outward unit normal to the boundary of the domain Ω.

### 4.2. Remarks on the Notion of Energy

The standard proof is correct, and gives one a tool to prove the desirable proposition concerning asymptotic stability. However, the terminology *energy norm* or *energy* for the quantity $E_{\mathrm{std}}$ defined via (Equation 10) is inappropriate and misleading for several reasons.

*First*, the quantity $E_{\mathrm{std}}$ does not even have the physical dimension of *physical* energy. *Second*, even if the physical dimension were corrected by a suitable constant multiplicative factor, the integral (Equation 10) would be different from the *physical* net total energy of the perturbation. Indeed, the *physical* net total energy is in the present case given by the formula
(11)$$E_{\mathrm{tot}}{=}_{\mathrm{def}}\int_{\Omega }\rho c_{V}\theta \;dv,$$
hence, the net total energy of the perturbation $\tilde{\theta }$ reduces to $\int_{\Omega }\rho c_{V}\tilde{\theta }\;dv$, which is different from (Equation 10). *Third*, the term “energy” for the quantity $E_{\mathrm{std}}$ is used even if one studies the stability of thermodynamically isolated system. However, in such a system the *physical* energy is a quantity that is *constant* in time, hence, it provides almost no clue concerning the evolution of the perturbation $\tilde{\theta }$. In particular, it can not be used for the characterisation of the *decay in time*.

Consequently, the quantity $E_{\mathrm{std}}$
*should not be referred to as the energy*. (At least when one wishes to understand the term energy as a term that has a physical meaning.) The proper term should be the mathematical one. Quantity $E_{\mathrm{std}}$ is, up to a constant multiplicative factor, the square of the norm of the perturbed temperature field $\tilde{\theta }$ in the Lebesgue space $L^{2}\left(\Omega \right)$.

Now, one is tempted to claim that the stability problem can not be solved by appealing to some physical concepts. Indeed, since the outlined proof is based on the *mathematical* concept of the norm in a Lebesgue space, one can argue that the true physical quantities such as the net total energy or the net entropy play no substantial role in the stability theory. (Note that the situation is different in hydrodynamic stability theory, see for example Serrin [8]. There, the Lebesgue space norm of the velocity perturbation ${∥\tilde{v}∥}_{L^{2}\left(\Omega \right)}$ is, up to a constant factor, tantamount to the physical *net kinetic energy* of the perturbation.) Consequently, the stability problem seems to be a purely mathematical problem that must be solved only by mathematically motivated manipulations with the governing equations.

As we shall demonstrate below, this is not the case. In fact, we show that thermodynamics plays a substantial role in nonlinear stability analysis. Moreover, we show that this is true even *in the case of thermodynamically open systems*.

### 4.3. Energy Method from the Perspective of Lyapunov Method

One can rephrase the outlined proof using the concept of the Lyapunov functional. The concept was introduced by Lyapunov [16] for the stability analysis of solutions to ordinary differential equations, see also La Salle and Lefschetz [17]. However, the concept works equally well for the stability analysis of solutions to partial differential equations, see for example Henry [18] or Flavin and Rionero [19]. (We do not discuss the relation between the Lyapunov functional and the norm in the corresponding function space, which is required for characterisation of the convergence $\tilde{\theta }\to 0$ as t→+∞. We are rather interested in finding a non-negative functional $V_{\mathrm{neq}}$ that vanishes if and only if the perturbation vanishes, and that decays along the trajectories predicted by the governing equations. The decay of the perturbation will be understood in the sense $V_{\mathrm{neq}}\left(\tilde{\theta }\right)\to 0$ as t→+∞. In this sense, we follow the practice introduced in Glansdorff and Prigogine [20]. In order to make this distinction visible, we refer to the Lyapunov functional constructed in this sense only as the Lyapunov *type* functional.)

Using the concept of Lyapunov functional, one can say that the square of Lebesgue norm ${∥\cdot ∥}_{L^{2}\left(\Omega \right)}$ of the perturbation $\tilde{\theta }$ is a natural Lyapunov functional characterising the stability of the equilibrium solution $\hat{\theta }$. Indeed, the functional $V_{\mathrm{std}}\left(\theta \right)$ defined as
(12)$$V_{\mathrm{std}}\left(\theta \right){=}_{\mathrm{def}}\int_{\Omega }\rho c_{V}{\left(\theta -\hat{\theta }\right)}^{2}\;dv,$$
is non-negative and it vanishes if and only if $\theta =\hat{\theta }$ in Ω, that is if and only if the steady equilibrium solution is attained. Further, the time derivative of the functional is negative along the trajectories determined by the corresponding governing Equation (7). This is easy to see if the temperature field θ is written in the form $\theta =\hat{\theta }+\tilde{\theta }$, which shows that the Lyapunov functional $V_{\mathrm{std}}$ is in fact identical, up to a constant coefficient, to the “energy” $E_{\mathrm{std}}$ of the perturbation as introduced in (Equation 10).

Now the question is the same. Is the choice of Lyapunov functional $V_{\mathrm{std}}$ motivated by a physical insight or is it just a matter of mathematical convenience?

## 5. Unconditional Asymptotic Stability: A Proof Based on Thermodynamical Concepts

### 5.1. Basic Facts from Thermodynamics of Continuous Media

Before proceeding with the thermodynamical analysis, let us recall some basic textbook facts from nonequilibrium continuum thermodynamics that are necessary for correct understanding of the physical background of the evolution Equation (Equation 4). The formulae below are straightforward generalisations of the standard formulae from classical equilibrium thermodynamics, see for example Callen [21], to the setting of spatially distributed fields. See for example Müller [22] for details.

#### 5.1.1. Specific Helmholtz Free Energy, Specific Entropy, Specific Internal Energy

First, if the rigid body of interest has a constant specific heat capacity at constant volume $c_{V}$, then the body can be characterised by the specific Helmholtz free energy ψ, $\left[\psi \right]=\frac{L^{2}}{T^{2}}$, in the form
(13)$$\psi {=}_{\mathrm{def}}-c_{V}\theta \left(\ln\left(\frac{\theta }{\theta_{\mathrm{ref}}}\right)-1\right),$$
where $\theta_{\mathrm{ref}}$ is a constant reference temperature value. Note that the specification of the Helmholtz free energy in fact determines how the body *stores the energy*, and this piece of information is usually the key starting point for modern theories of constitutive relations in continuum thermodynamics, see for example Rajagopal and Srinivasa [23] or Málek and Průša [24] for details. (The same holds also for the popular GENERIC framework, see Grmela and Öttinger [25], Öttinger and Grmela [26] and Pavelka et al. [27].) In particular, formulae for the specific Helmholtz free energy are known for many materials that are far more complex than the rigid heat conducting material, see for example Dressler et al. [28], Hron et al. [29], Málek et al. [30] or Málek et al. [31] for the case of polymeric liquids.

The formula for the specific entropy η, $\left[\eta \right]=\frac{L^{2}}{T^{2}\cdot Q}$, is obtained by differentiating the specific Helmholtz free energy ψ with respect to the temperature,
(14)$$\eta =-\frac{\partial \psi }{\partial \theta }.$$
In particular, for the specific Helmholtz free energy ψ in the form (Equation 13) we get
(15)$$\eta =c_{V}\ln\left(\frac{\theta }{\theta_{\mathrm{ref}}}\right).$$

The specific internal energy *e*, $\left[e\right]=\frac{L^{2}}{T^{2}}$, and the specific Helmholtz free energy ψ are related via Legendre transformation ψ=e−θη. This, in our simple case, yields
(16)$$e=c_{V}\theta .$$

#### 5.1.2. Entropy Production

Second, one needs to characterise the entropy production mechanisms in the body. Again, this piece of information is crucial in modern theory of constitutive relations in continuum thermodynamics, and entropy production mechanisms are known for many complex materials. In the present case, the entropy production is given by the formula $\xi =\frac{\zeta }{\theta }$, where
(17)$$\zeta {=}_{\mathrm{def}}\kappa \frac{{\left|\nabla \theta \right|}^{2}}{\theta }.$$
If (Equation 17) holds, then the energy flux $j_{e}$ in the body is given by the classical Fourier law
(18)$$j_{e}=-\kappa \nabla \theta ,$$
and the entropy flux $j_{\eta }$ is given by the standard formula $j_{\eta }=\frac{j_{e}}{\theta }$.

#### 5.1.3. Evolution Equations for the Total Energy, Specific Internal Energy and Specific Entropy

Finally, the evolution equations for the specific total energy $e_{\mathrm{tot}}=e+\frac{1}{2}{\left|v\right|}^{2}$, specific internal energy *e* and specific entropy η read, in the absence of external forces and heat sources, as follows
(19a)$$\begin{array}{c} \rho \frac{d}{dt}\left(e+\frac{1}{2}{\left|v\right|}^{2}\right)=\mathrm{div}\left(Tv-j_{e}\right), \end{array}$$
(19b)$$\begin{array}{c} \rho \frac{de}{dt}=T:D-\mathrm{div}j_{e}, \end{array}$$
(19c)$$\begin{array}{c} \rho \frac{d\eta }{dt}=\frac{\zeta }{\theta }-\mathrm{div}j_{\eta }, \end{array}$$
see for example Truesdell and Noll [32]. Here T denotes the Cauchy stress tensor, v denotes the velocity field, D denotes the symmetric part of the velocity gradient, and $\frac{d}{dt}$ denotes the material time derivative, that is for any quantity φ we have $\frac{d\varphi }{dt}{=}_{\mathrm{def}}\frac{\partial \varphi }{\partial t}+v•\nabla \varphi$. Symbol $\left|v\right|$ denotes the norm induced by the standard scalar product in $R^{3}$. In the case of heat conduction in a rigid body one has v=0, hence, T:D=0, and the material time derivative $\frac{d}{dt}$ coincides with the partial time derivative $\frac{\partial }{\partial t}$. The heat conduction Equation (Equation 4) is then obtained by the substitution of (Equation 16) and (Equation 18) into ([Disp-formula FD19b-entropy-21-00704]).

#### 5.1.4. Net Total Energy, Net Entropy

Having explicit formulae for the specific internal energy *e* and the specific entropy η, we can explicitly identify the *net total energy*
$E_{\mathrm{tot}}$ and *net entropy S* of the body occupying the domain Ω,
(20a)$$\begin{array}{c} E_{\mathrm{tot}}{=}_{\mathrm{def}}\int_{\Omega }\rho \left[\frac{1}{2}{\left|v\right|}^{2}+e\right]\;dv, \end{array}$$
(20b)$$\begin{array}{c} S{=}_{\mathrm{def}}\int_{\Omega }\rho \eta \;dv. \end{array}$$


Note that in the studied case of heat conduction in a rigid body the kinetic energy contribution $\int_{\Omega }\rho \frac{1}{2}{\left|v\right|}^{2}\;dv$ in ([Disp-formula FD20a-entropy-21-00704]) vanishes since we consider a fixed rigid body with v=0. Formula ([Disp-formula FD20a-entropy-21-00704]) however holds even for a moving continuous medium and it is written down for the sake of completeness. Since the concepts of the net total energy and net entropy are apparently well defined whenever one has an expression for the specific Helmholtz free energy, we see that these concepts are not exclusively restricted to the studied case of heat conduction in a fixed rigid body.

#### 5.1.5. Thermodynamically Isolated System

Once we have explicit formulae for the *energy flux* and the *entropy flux*, we know what boundary conditions imply that the system of interest is thermodynamically isolated. (Thermodynamically isolated system is a system that is not allowed to exchange mass and any form of energy with its surrounding.) The boundary condition that express the fact that the body is thermodynamically isolated is ${\left.\left(Tv-j_{e}\right)•n\right|}_{\partial \Omega }=0$, which in our setting translates to
(21)$${\left.\nabla \theta •n\right|}_{\partial \Omega }=0,$$
where n denotes the unit outward normal to Ω. Note that if the body is thermodynamically isolated then ([Disp-formula FD19a-entropy-21-00704]) implies that the net total energy is conserved, $\frac{dE_{\mathrm{tot}}}{dt}=0$.

### 5.2. Unconditional Asymptotic Stability of the Equilibrium Rest State in a Thermodynamically Isolated System

Now we are in the position to exploit thermodynamical concepts in nonlinear stability analysis. First, we investigate the stability of the spatially homogeneous equilibrium rest state in a *thermodynamically isolated* system. Then we proceed with the stability analysis of a steady state in a *thermodynamically open* setting. The stability problem for the spatially homogeneous equilibrium rest state is in fact a very simple problem, but it will motivate the techniques used in a more general setting. In both cases, we show that thermodynamical concepts can be used in a systematic construction of Lyapunov type functionals characterising the stability of the corresponding solution.

#### 5.2.1. Governing Equations for the Equilibrium Rest State

The *steady* solution $\hat{\theta }$ of (Equation 4) with the boundary condition (Equation 21) is a spatially homogeneous constant temperature field $\hat{\theta }=\theta_{\mathrm{bdr}}$. The value of $\theta_{\mathrm{bdr}}$ corresponds to the initial value of the net total energy $\hat{E_{\mathrm{tot}}}$, that is
(22)$$\theta_{\mathrm{bdr}}=\frac{\hat{E_{\mathrm{tot}}}}{\rho c_{V}\left|\Omega \right|},$$
where $\left|\Omega \right|$ denotes the volume of the domain occupied by the rigid body.

In other words, the equilibrium rest state temperature distribution in a thermodynamically isolated rigid body is spatially homogeneous. In particular, the temperature value inside the body corresponds to the temperature value on the boundary. Since $\hat{\theta }$ is a constant, we see that the associated *entropy production* given by (Equation 17) *vanishes*. This means that the temperature distribution $\hat{\theta }$ attained at the equilibrium rest state in the thermodynamically isolated body indeed deserves to be referred to as an *equilibrium* temperature distribution. Moreover, the physical notion of equilibrium (zero entropy production) coincides with the dynamical systems theory notion of equilibrium (right-hand side of (Equation 4) vanishes).

#### 5.2.2. Governing Equations for the Perturbation

We are interested in the stability of the equilibrium rest state $\hat{\theta }=\theta_{\mathrm{bdr}}$, which means that we need to solve the evolution equations
(23a)$$\begin{array}{c} \rho c_{V}\frac{\partial \theta }{\partial t}=\mathrm{div}\left(\kappa \nabla \theta \right), \end{array}$$
(23b)$$\begin{array}{c} {\left.\nabla \theta •n\right|}_{\partial \Omega }=0, \end{array}$$
(23c)$$\begin{array}{c} {\left.\theta \right|}_{t=0}=\theta_{\mathrm{init}}, \end{array}$$
and show that $\theta \to \hat{\theta }$ at t→+∈ for any initial spatially inhomogeneous temperature field $\theta_{\mathrm{init}}$. The initial temperature field $\theta_{\mathrm{init}}$ can be arbitrary, but it must satisfy some natural compatibility requirements. First, the initial temperature field $\theta_{\mathrm{init}}$ must be positive at every point of the domain. Second, the initial temperature field must be compatible with the given net total energy $\hat{E_{\mathrm{tot}}}$. (Net total energy must be conserved in thermodynamically isolated systems.) In other words, we require $E_{\mathrm{tot}}=\hat{E_{\mathrm{tot}}}$, which reduces to
(24)$$\int_{\Omega }\rho c_{V}\theta_{\mathrm{init}}\;dv=\int_{\Omega }\rho c_{V}\theta_{\mathrm{bdr}}\;dv.$$

#### 5.2.3. Construction of a Physically Motivated Lyapunov Functional—An Unsuccessful Attempt

When investigating the stability of the equilibrium steady state $\hat{\theta }$, we would like to identify a suitable Lyapunov functional. Before presenting a construction that actually works, it would be worthwhile to show a tempting construction that *does not work*. A natural physically motivated candidate for a Lyapunov type functional seems to be the (negative) net entropy *S*, since the net entropy *S* is in a thermodynamically isolated system a nondecreasing function of time. This is easy to see by integrating (19c) over the domain Ω, which yields
(25)$$\frac{dS}{dt}=\int_{\Omega }\frac{\zeta }{\theta }\;dv\geq 0,$$
where the *entropy flux*
$j_{\eta }$ vanishes in virtue of the boundary condition ([Disp-formula FD23b-entropy-21-00704]). The explicit formula for the net entropy functional *S* in our case reads
(26)$$S=\int_{\Omega }\rho c_{V}\ln\left(\frac{\theta }{\theta_{\mathrm{ref}}}\right)\;dv,$$
see (Equation 15) and ([Disp-formula FD20b-entropy-21-00704]), where the reference temperature $\theta_{\mathrm{ref}}$ can be, for the sake of convenience, fixed as $\theta_{\mathrm{ref}}=\theta_{\mathrm{bdr}}=\hat{\theta }$. Consequently, we see that $S(\hat{\theta }+\tilde{\theta })$ vanishes provided that $\tilde{\theta }=0$, which is a desirable property in the construction of a Lyapunov type functional.

However, the *net entropy functional does not provide sufficient information on the spatial distribution of the temperature*. In other words, $S(\hat{\theta }+\tilde{\theta })=0$ does not imply $\tilde{\theta }=0$, and, much worse, the functional can be both positive or negative depending on the particular choice of $\tilde{\theta }$. Consequently, the functional can not be used as a Lyapunov functional.

Does this mean that thermodynamics has nothing to say with respect to the construction of a Lyapunov type functional? Absolutely not. One has to recall that thermodynamics is based on two concepts—the *entropy* and the *energy*. One should not be dealt with in the absence of the other. Indeed, we can construct a suitable Lyapunov type functional if we use the energy in addition to the entropy.

#### 5.2.4. Construction of a Physically Motivated Lyapunov Type Functional—A Successful Attempt

We will exploit the famous formulation of the first and second law of thermodynamics by Clausius [33], namely the following statement:
The energy of the world is constant. The entropy of the world strives to a maximum.

In other words, the entropy of a *thermodynamically isolated system* attains *in the long run* the maximal possible value achievable at the given energy level. Note that although the original statement was formulated for spatially homogeneous systems, we can use it with a little modification also for spatially inhomogenenous systems. The only modification is that the energy and the entropy must be understood as the net total energy and the net entropy.

The maximum net entropy value achievable at the given net total energy level can be determined by solving a constrained maximisation problem. We want to maximise the *net entropy* (Equation 26) over all possible temperature fields θ that satisfy ([Disp-formula FD23b-entropy-21-00704]) and that have the *net total energy*
$E_{\mathrm{tot}}$ equal to the reference net total energy $\hat{E_{\mathrm{tot}}}$. This is easy to do using the Lagrange multiplier technique. The auxiliary functional for the constrained maximisation problem is
(27)$$L_{\Lambda }\left(\theta \right){=}_{\mathrm{def}}S-\Lambda (E_{\mathrm{tot}}-\hat{E_{\mathrm{tot}}}),$$
where Λ is the Lagrange multiplier. The Gâteaux derivative of $L_{\Lambda }\left(\theta \right)$ reads
(28)$$D_{}L_{\Lambda }\left(\theta \right)\left[\vartheta \right]=\int_{\Omega }\rho c_{V}\left(\frac{1}{\theta }-\Lambda \right)\vartheta \;dv.$$
Let us recall that the Gâteaux derivative $D_{}M\left(x\right)\left[y\right]$ of a functional M at point x in the direction y is defined as $D_{}M\left(x\right)\left[y\right]{=}_{\mathrm{def}}\lim_{s\to 0}\frac{M(x+sy)-M(x)}{s}$ which is tantamount to $D_{}M\left(x\right)\left[y\right]{=}_{\mathrm{def}}{\left.\frac{d}{ds}M\left(x+sy\right)\right|}_{s=0}$. If it is necessary to emphasize the variable against which we differentiate, we also write $D_{x}M\left(x\right)\left[y\right]$ instead of $D_{}M\left(x\right)\left[y\right]$. The derivative vanishes in all possible directions ϑ provided that $\Lambda =\frac{1}{\theta }$. The Lagrange multiplier Λ is a number, hence, the temperature field θ at which the derivative vanishes must be a spatially homogeneous temperature field. Using the constraint $E_{\mathrm{tot}}=\hat{E_{\mathrm{tot}}}$ we can therefore, conclude that the temperature field θ that for all ϑ satisfies
(29)$$D_{}L_{\Lambda }\left(\theta \right)\left[\vartheta \right]=0$$
is the uniform temperature field $\theta_{\mathrm{ref}}=\theta_{\mathrm{bdr}}=\hat{\theta }$. This confirms the expected fact that the *spatially homogeneous temperature field is the state that our thermodynamically isolated system wants to reach*.

Let us now exploit the fact that we know the value of the Lagrange multiplier Λ in (Equation 27), and let us investigate the functional
(30)$$L_{\frac{1}{\hat{\theta }}}\left(\theta \right){=}_{\mathrm{def}}S-\frac{1}{\hat{\theta }}(E_{\mathrm{tot}}-\hat{E_{\mathrm{tot}}}).$$
Explicit formula for the functional reads
(31)$$L_{\frac{1}{\hat{\theta }}}\left(\theta \right)=\int_{\Omega }\rho c_{V}\left[\ln\left(\frac{\theta }{\hat{\theta }}\right)-\frac{\theta }{\hat{\theta }}+1\right]\;dv.$$
Recall that the reference temperature has been chosen as $\theta_{\mathrm{ref}}=\hat{\theta }$. We observe that function
(32)$$f\left(\theta \right){=}_{\mathrm{def}}\ln\left(\frac{\theta }{\hat{\theta }}\right)-\frac{\theta }{\hat{\theta }}+1$$
is for θ>0
*negative* whenever $\theta \neq \hat{\theta }$, and it vanishes if and only if $\theta =\hat{\theta }$. Further, this function is a concave function. The plot of the function *f* is shown in Figure 1a.

Since the temperature field θ that solves (23) remains positive, we see that the functional $L_{\frac{1}{\hat{\theta }}}\left(\theta \right)$ is non-positive for all possible solutions to (23). Moreover, it vanishes if and only if $\theta =\hat{\theta }$ everywhere in the domain Ω. In other words it vanishes at the equilibrium rest state $\hat{\theta }$, and it provides a *control on the spatial variations of the temperature field* with respect to the equilibrium value. This means that the functional
(33)$$V_{\mathrm{eq}}\left(\theta \right){=}_{\mathrm{def}}-L_{\frac{1}{\hat{\theta }}}\left(\theta \right),$$
that is
(34)$$V_{\mathrm{eq}}\left(\theta \right)=-\int_{\Omega }\rho c_{V}\left[\ln\left(\frac{\theta }{\hat{\theta }}\right)-\frac{\theta }{\hat{\theta }}+1\right]\;dv,$$
is a good candidate for a Lyapunov type functional characterising the stability of the equilibrium rest state $\hat{\theta }$.

It remains to check that the time derivative of the proposed Lyapunov type functional $V_{\mathrm{eq}}$ is non-positive provided that the temperature field evolves according to the governing Equation (23). First, we observe that in a thermodynamically isolated system we get
(35)$$\frac{dE_{\mathrm{tot}}}{dt}=0.$$
This follows by the integration of ([Disp-formula FD19a-entropy-21-00704]) over the domain Ω, and from the fact that energy flux $j_{e}$ vanishes on the boundary. Second, the entropy of the thermodynamically isolated system is a nondecreasing function, see (Equation 25). Consequently, we see that
(36)$$\frac{dV_{\mathrm{eq}}}{dt}=-\frac{d}{dt}\left[S-\frac{1}{\hat{\theta }}(E_{\mathrm{tot}}-\hat{E_{\mathrm{tot}}})\right]=-\frac{dS}{dt}=-\int_{\Omega }\frac{\zeta }{\theta }\;dv\leq 0.$$
Moreover, the derivative vanishes if and only if the given temperature field is spatially homogeneous. (Recall that $\zeta =\kappa \frac{{\left|\nabla \theta \right|}^{2}}{\theta }$, see (Equation 17).) This concludes that $V_{\mathrm{eq}}$ is indeed a suitable Lyapunov type functional characterising the stability of the equilibrium rest state $\hat{\theta }$, hence, the equilibrium rest state is unconditionally asymptotically stable.

The fact that the functional of the type $S-\frac{1}{\theta_{\mathrm{bdr}}}E_{\mathrm{tot}}$ can be used as a Lyapunov functional characterising the stability of the equilibrium rest state in a thermodynamically isolated system is well known, see Coleman [1], Gurtin [2], Šilhavý [34], Ericksen [35] or Grmela and Öttinger [25]. In fact Gurtin [2] attributes this observation to Duhem [36], yet the core idea can be, for spatially homogeneous systems, found already in the works of Clausius [33] and Gibbs [37], Gibbs [38].

Interestingly, the functional $S-\frac{1}{\theta_{\mathrm{bdr}}}E_{\mathrm{tot}}$ is not used or even mentioned in standard treatises on nonlinear stability analysis, see Joseph [34] or Straughan [5]. This is in a sense natural, since these works are mostly focused on thermodynamically *open* systems, where the approach introduced in the seminal work by Coleman [1] is largely inapplicable. On the other hand, this omission can be seen as an evidence of the perceived inapplicability of genuine thermodynamical concepts in the nonlinear stability analysis of thermodynamically open systems.

#### 5.2.5. Relation to the Standard Energy Method

The constructed functional $V_{\mathrm{eq}}$ coincides, up to a constant, with the standard functional $V_{\mathrm{std}}$ introduced in Section 4 provided that the temperature perturbation is small. Indeed, if $\frac{\theta }{\theta_{\mathrm{ref}}}\approx 1$, then $f\left(\theta \right)\approx \frac{\theta^{2}}{\theta_{\mathrm{ref}}^{2}}$, and consequently $V_{\mathrm{eq}}\approx V_{\mathrm{std}}$.

Note also that the functional $V_{\mathrm{eq}}$ can be seen, up to a constant factor, as the generalisation of the classical concept of *exergy*/*available energy*, see for example Bruges [39], to spatially inhomogeneous systems. Moreover a variant of the functional $V_{\mathrm{eq}}$, namely the functional $E_{\mathrm{tot}}-\hat{\theta }S$ is also used in the engineering practice in the so-called entropy generation analysis, see for example Sciacovelli et al. [40].

### 5.3. Unconditional Asymptotic Stability of a General Steady State in a Thermodynamically Open System

Having identified a physically motivated Lyapunov functional for the stability analysis of the rest state in a thermodynamically *isolated* system, we can proceed with the stability analysis of steady solution in a thermodynamically *open* system.

#### 5.3.1. Governing Equations for the Non-Equilibrium Steady State

We consider the heat conduction problem in a rigid body with a given temperature value $\theta_{\mathrm{bdr}}$ on the boundary, where the temperature value $\theta_{\mathrm{bdr}}$ on the boundary can be *position dependent*. (A part of the boundary can be kept at a different temperature than the other. A good model problem is the heat conduction problem in a rod that has its ends kept at different temperatures.) This means that the analysis below *is not restricted* to the setting of body “immersed in a environment of [spatially uniform] temperature”, see Coleman [1] and similar works such as Gurtin [41] and Gurtin [2].

Let $\hat{\theta }$ again denotes the steady solution to the boundary-value problem (5), that is the temperature field $\hat{\theta }$ solves the problem
(37a)$$\begin{array}{c} 0=\mathrm{div}(\kappa \nabla \hat{\theta }), \end{array}$$
(37b)$$\begin{array}{c} {\left.\hat{\theta }\right|}_{\partial \Omega }=\theta_{\mathrm{bdr}}. \end{array}$$
This is the steady solution whose stability we want to investigate.

Note that if the boundary condition $\theta_{\mathrm{bdr}}$ is *spatially inhomogeneous*, then the solution $\hat{\theta }$ is spatially inhomogeneous as well. Consequently, the entropy production (Equation 17) at the steady state $\hat{\theta }$ is positive, which makes the widely used mathematical term *equilibrium solution* for $\hat{\theta }$ a bit problematic from the physical point of view. The system we are interested in is from the thermodynamical point of view *out of thermodynamical equilibrium*. (Entropy is being produced.) Therefore $\hat{\theta }$ is from this point of view a *non-equilibrium steady state*.

#### 5.3.2. Governing Equations for the Perturbation

The evolution of the perturbed temperature field $\theta =\hat{\theta }+\tilde{\theta }$ is governed by the Equation (7), where $\theta_{\mathrm{init}}$ is an arbitrary initial condition, while the steady state temperature field $\hat{\theta }$ is the solution to (37). Consequently, the time evolution of the perturbation $\tilde{\theta }$ is governed by
(38a)$$\begin{array}{c} \rho c_{V}\frac{\partial \tilde{\theta }}{\partial t}=\mathrm{div}(\kappa \nabla \tilde{\theta }), \end{array}$$
(38b)$$\begin{array}{c} {\left.\tilde{\theta }\right|}_{\partial \Omega }=0, \end{array}$$
(38c)$$\begin{array}{c} {\left.\tilde{\theta }\right|}_{t=0}=\theta_{\mathrm{init}}-\hat{\theta }. \end{array}$$
The aim is to show that the perturbation $\tilde{\theta }$ vanishes as time goes to infinity irrespective of the choice of the initial condition $\theta_{\mathrm{init}}$.

#### 5.3.3. Heuristics Concerning the Construction of a Lyapunov Functional

Concerning the stability analysis of the steady state $\hat{\theta }$ we again want to exploit the concept of Lyapunov functional. The following observation will be helpful. Let us assume that we have a quadratic positive definite functional defined on the real line, say
(39)$$V_{\mathrm{eq}}\left({\tilde{x}}_{\mathrm{eq}}∥{\hat{x}}_{\mathrm{eq}}\right){=}_{\mathrm{def}}{\tilde{x}}_{\mathrm{eq}}^{2},$$
that can be used as the Lyapunov functional for the stability analysis of an equilibrium rest state ${\hat{x}}_{\mathrm{eq}}$. Here the perturbation with respect to the rest state is denoted as ${\tilde{x}}_{\mathrm{eq}}$, and the complete perturbed field *x* is defined as
(40)$$x={\hat{x}}_{\mathrm{eq}}+{\tilde{x}}_{\mathrm{eq}}.$$
Note that in terms of the complete perturbed field *x* we get $V_{\mathrm{eq}}\left({\tilde{x}}_{\mathrm{eq}}∥{\hat{x}}_{\mathrm{eq}}\right)={\left(x-{\hat{x}}_{\mathrm{eq}}\right)}^{2}$. Consequently, we also use, whenever appropriate, the notation $V_{\mathrm{eq}}\left(x\right){=}_{\mathrm{def}}{\left(x-{\hat{x}}_{\mathrm{eq}}\right)}^{2}$ or
(41)$$V_{\mathrm{eq}}\left(x\right){=}_{\mathrm{def}}V_{\mathrm{eq}}\left({\tilde{x}}_{\mathrm{eq}}∥{\hat{x}}_{\mathrm{eq}}\right).$$
The latter notation $V_{\mathrm{eq}}\left(x\right)$ indicates that we are dealing with the complete perturbed field *x*, while the former notation $V_{\mathrm{eq}}\left({\tilde{x}}_{\mathrm{eq}}∥{\hat{x}}_{\mathrm{eq}}\right)$ indicates that we are interested in the stability of the steady state ${\hat{x}}_{\mathrm{eq}}$ subject to perturbations ${\tilde{x}}_{\mathrm{eq}}$.

Now we want to construct a new functional $V_{\mathrm{neq}}\left({\tilde{x}}_{\mathrm{neq}}∥{\hat{x}}_{\mathrm{neq}}\right)$ that could serve as a Lyapunov functional characterising the stability of the non-equilibrium steady state ${\hat{x}}_{\mathrm{neq}}$. The point ${\hat{x}}_{\mathrm{neq}}$ represents the non-equilibrium steady state whose stability we are interested in, and ${\tilde{x}}_{\mathrm{neq}}$
*denotes the perturbation with respect to the nonequlibrium state*
${\hat{x}}_{\mathrm{neq}}$. The complete perturbed field *x* is again composed of the perturbation ${\tilde{x}}_{\mathrm{neq}}$ and the non-equilibrium steady state ${\hat{x}}_{\mathrm{neq}}$,
(42)$$x={\hat{x}}_{\mathrm{neq}}+{\tilde{x}}_{\mathrm{neq}}.$$
We want the new functional $V_{\mathrm{neq}}$ to vanish if the perturbation ${\tilde{x}}_{\mathrm{neq}}$ vanishes, and to be positive otherwise.

The new functional $V_{\mathrm{neq}}$ can be constructed from $V_{\mathrm{eq}}$ as follows. We “subtract” the tangent to the graph of the former functional $V_{\mathrm{eq}}$ at the point ${\hat{x}}_{\mathrm{neq}}$ from the graph of the former functional $V_{\mathrm{eq}}$. (See Figure 2 for a sketch of the construction.) In other words, the new functional $V_{\mathrm{neq}}$ is defined as
(43a)$$V_{\mathrm{neq}}\left({\tilde{x}}_{\mathrm{neq}}∥{\hat{x}}_{\mathrm{neq}}\right){=}_{\mathrm{def}}V_{\mathrm{eq}}\left({\hat{x}}_{\mathrm{neq}}+{\tilde{x}}_{\mathrm{neq}}-{\hat{x}}_{\mathrm{eq}}∥{\hat{x}}_{\mathrm{eq}}\right)-V_{\mathrm{eq}}\left({\hat{x}}_{\mathrm{neq}}-{\hat{x}}_{\mathrm{eq}}∥{\hat{x}}_{\mathrm{eq}}\right)-{\left.\frac{dV_{\mathrm{eq}}}{dx}\right|}_{x={\hat{x}}_{\mathrm{neq}}}{\tilde{x}}_{\mathrm{neq}},$$
or in other words as
(43b)$$V_{\mathrm{neq}}\left({\tilde{x}}_{\mathrm{neq}}∥{\hat{x}}_{\mathrm{neq}}\right){=}_{\mathrm{def}}V_{\mathrm{eq}}\left({\hat{x}}_{\mathrm{neq}}+{\tilde{x}}_{\mathrm{neq}}\right)-V_{\mathrm{eq}}\left({\hat{x}}_{\mathrm{neq}}\right)-{\left.\frac{dV_{\mathrm{eq}}}{dx}\right|}_{x={\hat{x}}_{\mathrm{neq}}}{\tilde{x}}_{\mathrm{neq}}.$$
Formula (43) can be as well read as the “remainder” after subtracting the first order expansion of the original functional $V_{\mathrm{eq}}$ at the point ${\hat{x}}_{\mathrm{neq}}$ from the functional $V_{\mathrm{eq}}$.

In the case of functional (Equation 39) we get
(44)$$V_{\mathrm{neq}}\left({\tilde{x}}_{\mathrm{neq}}∥{\hat{x}}_{\mathrm{neq}}\right)={\left({\hat{x}}_{\mathrm{neq}}+{\tilde{x}}_{\mathrm{neq}}-{\hat{x}}_{\mathrm{eq}}\right)}^{2}-{\left({\hat{x}}_{\mathrm{neq}}-{\hat{x}}_{\mathrm{eq}}\right)}^{2}-2\left({\hat{x}}_{\mathrm{neq}}-{\hat{x}}_{\mathrm{eq}}\right){\tilde{x}}_{\mathrm{neq}}={\tilde{x}}_{\mathrm{neq}}^{2}.$$
The newly constructed functional $V_{\mathrm{neq}}\left({\tilde{x}}_{\mathrm{neq}}∥{\hat{x}}_{\mathrm{neq}}\right)$ is positive provided ${\tilde{x}}_{\mathrm{neq}}\neq 0$. Moreover, it vanishes at ${\tilde{x}}_{\mathrm{neq}}=0$, that is if the state of the system $x={\hat{x}}_{\mathrm{neq}}+{\tilde{x}}_{\mathrm{neq}}$ is identical to the chosen non-equilibrium steady state ${\hat{x}}_{\mathrm{neq}}$. Consequently, $V_{\mathrm{neq}}\left({\tilde{x}}_{\mathrm{neq}}∥{\hat{x}}_{\mathrm{neq}}\right)$ is a reasonable *guess* concerning the Lyapunov functional suitable for the stability analysis of the non-equilibrium state ${\hat{x}}_{\mathrm{neq}}$.

It remains to show that the newly constructed functional is decreasing along the trajectories predicted by the corresponding governing equations for *x*. If this can be shown, then the newly constructed functional is indeed a Lyapunov functional suitable for the analysis of the stability of the non-equilibrium state ${\hat{x}}_{\mathrm{neq}}$. In this heuristic argument we however do not consider any underlying dynamical system, hence, we can not proceed further in the study of the property $\frac{dV_{\mathrm{neq}}\left({\tilde{x}}_{\mathrm{neq}}∥{\hat{x}}_{\mathrm{neq}}\right)}{dt}\leq 0$.

We note that the outlined construction is quite general, and it can be easily extended to the multidimensional or even infinite-dimensional setting. The key property that guarantees a meaningful outcome of the outlined construction of $V_{\mathrm{neq}}$ is the convexity of the functional characterising the stability of the equilibrium rest state $V_{\mathrm{eq}}$. The origins of the outlined construction can be, to our best knowledge, traced back to Ericksen [42].

#### 5.3.4. Construction of a Physically Motivated Lyapunov Functional—General Remarks

Let us now follow the outlined heuristic in the case of dynamical systems in continuum thermodynamics, and especially in the case of heat conduction. The Lyapunov functional $V_{\mathrm{eq}}$ for the *equilibrium rest state* in a thermodynamically closed system has been identified in (Equation 33), and it is given by the formula
(45)$$V_{\mathrm{eq}}=-\left\{S-\frac{1}{\hat{\theta }}(E_{\mathrm{tot}}-\hat{E_{\mathrm{tot}}})\right\}.$$
However, if we want to use (Equation 45) as a building block for a Lyapunov functional characterising the stability of a steady non-equilibrium state with a spatially inhomogeneous temperature field $\hat{\theta }$, we see that (Equation 45) does not define a functional. It does not assign a *real number* to the given state of the system (temperature field). (While *S* and $E_{\mathrm{tot}}$ in (Equation 45) are numbers even if one deals with spatially inhomogeneous temperature field, the factor $\frac{1}{\hat{\theta }}$ is in the spatially inhomogenous setting a position dependent function.) This can be fixed if we realise that the Lyapunov functional characterising the stability of the *equilibrium rest state* can be rewritten as
(46)$$V_{\mathrm{eq}}=-\frac{1}{\hat{\theta }}\left\{\hat{\theta }S-(E_{\mathrm{tot}}-\hat{E_{\mathrm{tot}}})\right\}=-\frac{1}{\hat{\theta }}\left\{\int_{\Omega }\hat{\theta }\rho \eta \;dv-\left(\int_{\Omega }\rho \left[\frac{1}{2}{\left|v\right|}^{2}+e\right]\;dv-\hat{E_{\mathrm{tot}}}\right)\right\},$$
where we have used the definition of the net total energy $E_{\mathrm{tot}}$ and the net entropy *S*, see (20). The factor $\frac{1}{\hat{\theta }}$ in (Equation 46) is immaterial in the stability analysis of a spatially homogeneous equilibrium rest state $\hat{\theta }$. Indeed the modified Lyapunov functional
(47)$$V_{\mathrm{meq}}{=}_{\mathrm{def}}-\left\{\int_{\Omega }\hat{\theta }\rho \eta \;dv-\left(\int_{\Omega }\rho \left[\frac{1}{2}{\left|v\right|}^{2}+e\right]\;dv-\hat{E_{\mathrm{tot}}}\right)\right\}$$
can serve as well as the original Lyapunov functional (Equation 45) in the stability analysis of the spatially homogeneous equilibrium rest state.

Introducing the notation
(48)$$S_{\hat{\theta }}{=}_{\mathrm{def}}\int_{\Omega }\rho \hat{\theta }\eta \;dv,$$
we see that (Equation 47) can be rewritten as
(49)$$V_{\mathrm{meq}}=-\left\{S_{\hat{\theta }}-(E_{\mathrm{tot}}-\hat{E_{\mathrm{tot}}})\right\}.$$
Note that the definition (Equation 49) of $V_{\mathrm{meq}}$ is general enough to be applicable whenever one deals with a continuous medium with a well defined specific Helmholtz free energy. It is by no means restricted to the specific problem of heat conduction in a rigid body.

The benefit of using $V_{\mathrm{meq}}$ instead of $V_{\mathrm{eq}}$ lies in the fact that the temperature field $\hat{\theta }$ is now placed under the integration sign, hence, $V_{\mathrm{meq}}$ defines a functional even if $\hat{\theta }$ is a spatially inhomogeneous temperature field. This subtlety does not matter if $\hat{\theta }$ is a constant temperature field. On the other hand, if $\hat{\theta }$ is spatially inhomogeneous, it allows us to use the Lyapunov functional $V_{\mathrm{meq}}$ as functional that serves as a building block in the construction of the Lyapunov functional $V_{\mathrm{neq}}$ for the non-equlibrium steady state $\hat{\theta }$.

#### 5.3.5. Construction of a Physically Motivated Lyapunov Type Functional—Heat Conduction in a Rigid Body

Now we are in a position to follow the construction outlined in Section 5.3.3 and Section 5.3.4 in the specific case of heat conduction in a rigid body. (Governing equations for the perturbation are the equations (23).) Let W denote the vector of state variables, and let $\tilde{W}$ and $\hat{W}$ denote a perturbation and a non-equilibrium steady state respectively. The candidate for Lyapunov type functional is defined as
(50a)$$V_{\mathrm{neq}}(\tilde{W}∥\hat{W}){=}_{\mathrm{def}}-\left\{S_{\hat{\theta }}\left(\tilde{W}∥\hat{W}\right)-E\left(\tilde{W}∥\hat{W}\right)\right\},$$
where
(50b)$$\begin{array}{c} S_{\hat{\theta }}\left(\tilde{W}∥\hat{W}\right){=}_{\mathrm{def}}S_{\hat{\theta }}(\hat{W}+\tilde{W})-S_{\hat{\theta }}(\hat{W})-{\left.D_{W}S_{\hat{\theta }}(W)\right|}_{W=\hat{W}}\left[\tilde{W}\right], \end{array}$$
(50c)$$\begin{array}{c} E(\tilde{W}∥\hat{W}){=}_{\mathrm{def}}E_{\mathrm{tot}}(\hat{W}+\tilde{W})-E_{\mathrm{tot}}(\hat{W})-{\left.D_{W}E_{\mathrm{tot}}(W)\right|}_{W=\hat{W}}\left[\tilde{W}\right], \end{array}$$
and the functionals $S_{\hat{\theta }}\left(W\right)$ and $E_{\mathrm{tot}}\left(W\right)$ are defined as
(51a)$$\begin{array}{cc} S_{\hat{\theta }}\left(W\right) & {=}_{\mathrm{def}}\int_{\Omega }\rho \hat{\theta }\eta \left(W\right)\;dv, \end{array}$$
(51b)$$\begin{array}{cc} E_{\mathrm{tot}}\left(W\right) & {=}_{\mathrm{def}}\int_{\Omega }\rho e\left(W\right)\;dv. \end{array}$$
In (50) we have rewritten (43) in the infinite-dimensional setting, meaning that the derivative has been replaced by the Gâteaux derivative.

In our case the specific entropy η and the specific internal energy *e* are given by the formulae
(52)$$\eta \left(W\right)=c_{V}\ln\left(\frac{\theta }{\theta_{\mathrm{ref}}}\right),\;e\left(W\right)=c_{V}\theta ,$$
see (Equation 15) and (Equation 16), and the only state variable is the temperature field, that is $W{=}_{\mathrm{def}}\theta$, $\hat{W}{=}_{\mathrm{def}}\hat{\theta }$ and $\tilde{W}{=}_{\mathrm{def}}\tilde{\theta }$. The Gâteaux derivatives of the functionals $S_{\hat{\theta }}\left(W\right)$ and $E_{\mathrm{tot}}\left(W\right)$ read
(53)$${\left.D_{W}S_{\hat{\theta }}\left(W\right)\right|}_{W=\hat{W}}\left[\tilde{W}\right]=\int_{\Omega }\rho c_{V}\tilde{\theta }\;dv,\;{\left.D_{W}E_{\mathrm{tot}}\left(W\right)\right|}_{W=\hat{W}}\left[\tilde{W}\right]=\int_{\Omega }\rho c_{V}\tilde{\theta }\;dv.$$
Note that the differentiation in (Equation 53) requires one to vary only W. The factor $\hat{\theta }$ in $S_{\hat{\theta }}\left(W\right)$ is left intact although we are differentiating with respect to the temperature. This calculation reveals that
(54)$$E\left(\tilde{W}∥\hat{W}\right)=0,\;-S_{\hat{\theta }}\left(\tilde{W}∥\hat{W}\right)=\int_{\Omega }\rho c_{V}\hat{\theta }\left[\frac{\tilde{\theta }}{\hat{\theta }}-\ln\left(1+\frac{\tilde{\theta }}{\hat{\theta }}\right)\right]\;dv.$$
The function under the integration sign in $S_{\hat{\theta }}\left(\tilde{W}∥\hat{W}\right)$ is
(55)$$g\left(\tilde{\theta }\right){=}_{\mathrm{def}}\left[\frac{\tilde{\theta }}{\hat{\theta }}-\ln\left(1+\frac{\tilde{\theta }}{\hat{\theta }}\right)\right],$$
see the plot shown in Figure 1b, and it is well defined for $\tilde{\theta }>-\hat{\theta }$. The governing Equation (7) guarantees that the temperature field $\theta =\tilde{\theta }+\hat{\theta }$ remains positive provided that the initial temperature field $\theta_{\mathrm{init}}$ is positive, see for example Friedman [43], Ladyzhenskaya et al. [44] or Lieberman [45]. The pointwise values of the temperature perturbation $\tilde{\theta }$ therefore, remain in the interval $\left(-\hat{\theta },+\infty \right)$, and the function *g* remains well defined for any temperature field predicted by the corresponding governing equation. Moreover, the function *g*, and hence, the integrand in the expression for $S_{\hat{\theta }}\left(\tilde{W}∥\hat{W}\right)$, is positive whenever $\tilde{\theta }\neq 0$, and it vanishes at $\tilde{\theta }=0$.

This implies that the functional $V_{\mathrm{neq}}$ given by the explicit formula
(56)$$V_{\mathrm{neq}}(\tilde{W}∥\hat{W})=\int_{\Omega }\rho c_{V}\hat{\theta }\left[\frac{\tilde{\theta }}{\hat{\theta }}-\ln\left(1+\frac{\tilde{\theta }}{\hat{\theta }}\right)\right]\;dv$$
is well defined and non-negative for any achievable temperature field $\hat{\theta }+\tilde{\theta }$, and it vanishes if and only if $\tilde{\theta }=0$ in the whole domain Ω. Therefore, we can conclude that the functional $V_{\mathrm{neq}}$ is a good candidate for a Lyapunov type functional characterising the stability of non-equilibrium steady state $\hat{\theta }$. In particular, it provides a control on the *spatial inhomogeneity* of the perturbation $\tilde{\theta }$.

#### 5.3.6. Time Derivative of the Lyapunov Type Functional

It remains to show that the time derivative is non-positive,$\frac{dV_{\mathrm{neq}}}{dt}\leq 0$. The time derivative must be evaluated with the help of the governing equations (38) for the perturbation $\tilde{\theta }$. The key difficulty in evaluating the time derivative is that the heat flux *does not vanish on the boundary* since we are dealing with a thermodynamically *open* system. In particular, we can not exploit the equalities ${\left.\nabla \hat{\theta }•n\right|}_{\partial \Omega }=0$ or ${\left.\nabla \tilde{\theta }•n\right|}_{\partial \Omega }=0$ as in the case of a thermodynamically *isolated* system. In our case (thermodynamically open system), the boundary condition reads ${\left.\tilde{\theta }\right|}_{\partial \Omega }=0,$ which in general means that ${\left.\nabla \tilde{\theta }•n\right|}_{\partial \Omega }\neq 0$.

In calculating the time derivative $\frac{dV_{\mathrm{neq}}}{dt}$, we can either directly differentiate formula (Equation 56), or we can try to exploit the fact that $V_{\mathrm{neq}}$ is given by (50), and use formulae for the time derivatives of the net total energy $E_{\mathrm{tot}}$ and the net entropy *S*.

The direct differentiation of $V_{\mathrm{neq}}$ would give
(57)$$\begin{array}{r} \frac{dV_{\mathrm{neq}}}{dt}=\frac{d}{dt}\int_{\Omega }\rho c_{V}\hat{\theta }\left[\frac{\tilde{\theta }}{\hat{\theta }}-\ln\left(1+\frac{\tilde{\theta }}{\hat{\theta }}\right)\right]\;dv=\int_{\Omega }\left(\rho c_{V}\frac{\partial \tilde{\theta }}{\partial t}-\rho c_{V}\frac{1}{1+\frac{\tilde{\theta }}{\hat{\theta }}}\frac{\partial \tilde{\theta }}{\partial t}\right)\;dv=\int_{\Omega }\left(\mathrm{div}\left(\kappa \nabla \tilde{\theta }\right)-\frac{1}{1+\frac{\tilde{\theta }}{\hat{\theta }}}\mathrm{div}\left(\kappa \nabla \tilde{\theta }\right)\right)\;dv\\ =\int_{\Omega }\left(\mathrm{div}\left(\kappa \nabla \left[\hat{\theta }+\tilde{\theta }\right]\right)-\frac{1}{1+\frac{\tilde{\theta }}{\hat{\theta }}}\mathrm{div}\left(\kappa \nabla \left[\hat{\theta }+\tilde{\theta }\right]\right)\right)\;dv=\int_{\Omega }\left(\mathrm{div}\left(\left[1-\frac{1}{1+\frac{\tilde{\theta }}{\hat{\theta }}}\right]\kappa \nabla \left[\hat{\theta }+\tilde{\theta }\right]\right)+\kappa \nabla \frac{\hat{\theta }}{\hat{\theta }+\tilde{\theta }}•\nabla \left[\hat{\theta }+\tilde{\theta }\right]\right)\;dv\\ =\int_{\Omega }\mathrm{div}\left(\left[1-\frac{1}{1+\frac{\tilde{\theta }}{\hat{\theta }}}\right]\kappa \nabla \left[\hat{\theta }+\tilde{\theta }\right]\right)\;dv+\int_{\Omega }\kappa \nabla \frac{1}{1+\frac{\tilde{\theta }}{\hat{\theta }}}•\nabla \left[\hat{\theta }\left(1+\frac{\tilde{\theta }}{\hat{\theta }}\right)\right]\;dv\\ =\int_{\Omega }\mathrm{div}\left(\left[1-\frac{1}{1+\frac{\tilde{\theta }}{\hat{\theta }}}\right]\kappa \nabla \left[\hat{\theta }+\tilde{\theta }\right]\right)\;dv+\int_{\Omega }\kappa \nabla \hat{\theta }•\nabla \ln\left(1+\frac{\tilde{\theta }}{\hat{\theta }}\right)\;dv-\int_{\Omega }\kappa \hat{\theta }\nabla \ln\left(1+\frac{\tilde{\theta }}{\hat{\theta }}\right)•\nabla \ln\left(1+\frac{\tilde{\theta }}{\hat{\theta }}\right)\;dv\\ =\int_{\Omega }\mathrm{div}\left(\left[1-\frac{1}{1+\frac{\tilde{\theta }}{\hat{\theta }}}\right]\kappa \nabla \left[\hat{\theta }+\tilde{\theta }\right]\right)\;dv+\int_{\Omega }\mathrm{div}\left(\kappa \left[\nabla \hat{\theta }\right]\ln\left(1+\frac{\tilde{\theta }}{\hat{\theta }}\right)\right)\;dv-\int_{\Omega }\kappa \hat{\theta }\nabla \ln\left(1+\frac{\tilde{\theta }}{\hat{\theta }}\right)•\nabla \ln\left(1+\frac{\tilde{\theta }}{\hat{\theta }}\right)\;dv, \end{array}$$
where we have exploited the evolution equation for the temperature perturbation (38) and the fact that $\hat{\theta }$ solves (37), that is $\mathrm{div}(\kappa \nabla \hat{\theta })=0$. Now it remains to use the Stokes theorem and boundary condition ([Disp-formula FD38b-entropy-21-00704]), that is ${\left.\tilde{\theta }\right|}_{\partial \Omega }=0$, which yields
(58)$$\frac{dV_{\mathrm{neq}}}{dt}=-\int_{\Omega }\kappa \hat{\theta }\nabla \ln\left(1+\frac{\tilde{\theta }}{\hat{\theta }}\right)•\nabla \ln\left(1+\frac{\tilde{\theta }}{\hat{\theta }}\right)\;dv.$$
This is the same result as that we report in Equation (Equation 71). Although the direct differentiation is a legitimate technique, it is however a brute force approach.

Namely, the presence of the logarithm term on the right hand side of (Equation 58) seems to be a consequence of a purely formal manipulation. Therefore we would prefer the procedure outlined on the following lines, which shows that the logarithmic term naturally appears in the *entropy production*, and hence, also in the final formula for the time derivative. This approach, unlike the direct differentiation, helps us to keep track of the physical origin of the terms in the time derivative.

Since the Lyapunov type functional $V_{\mathrm{neq}}$ includes the term (Equation 48) that is defined in terms of the specific entropy, we need to first formulate governing equations for the relative specific entropy $\tilde{\eta }{=}_{\mathrm{def}}\eta -\hat{\eta }$, that is
(59)$$\tilde{\eta }{=}_{\mathrm{def}}\eta \left(\hat{W}+\tilde{W}\right)-\eta \left(\hat{W}\right).$$
This quantity measures the difference between the specific entropy at the perturbed state $\eta (\hat{W}+\tilde{W})$ and the specific entropy at the non-equilibrium steady state $\eta (\hat{W})$. In our case, the explicit formula for $\tilde{\eta }$ in terms of temperature reads
(60)$$\tilde{\eta }=c_{V}\ln\left(1+\frac{\tilde{\theta }}{\hat{\theta }}\right),$$
and the evolution equation for $\tilde{\eta }$ is
(61)$$\rho \frac{\partial \tilde{\eta }}{\partial t}=\frac{\kappa }{c_{V}^{2}}\nabla \tilde{\eta }•\nabla \tilde{\eta }+\frac{2\kappa }{c_{V}^{2}}\nabla \tilde{\eta }•\nabla \hat{\eta }+\mathrm{div}\left(\frac{\kappa }{c_{V}}\nabla \tilde{\eta }\right).$$
Equation (Equation 61) follows via subtracting the equations
(62a)$$\begin{array}{c} \rho \frac{\partial }{\partial t}\left(\hat{\eta }+\tilde{\eta }\right)=\kappa \frac{{\left|\nabla (\hat{\theta }+\tilde{\theta })\right|}^{2}}{{\left(\hat{\theta }+\tilde{\theta }\right)}^{2}}+\mathrm{div}\left\{\kappa \frac{\nabla (\hat{\theta }+\tilde{\theta })}{\hat{\theta }+\tilde{\theta }}\right\}, \end{array}$$
(62b)$$\begin{array}{c} \rho \frac{\partial \hat{\eta }}{\partial t}=\kappa \frac{{\left|\nabla \hat{\theta }\right|}^{2}}{{\hat{\theta }}^{2}}+\mathrm{div}\left\{\kappa \frac{\nabla \hat{\theta }}{\hat{\theta }}\right\}, \end{array}$$
where ([Disp-formula FD62b-entropy-21-00704]) is the entropy evolution Equation (19c) formulated for the non-equilibrium steady state $\eta =\hat{\eta }=\eta \left(\hat{W}\right)$, and ([Disp-formula FD62a-entropy-21-00704]) is the entropy evolution Equation (19c) formulated for the perturbed entropy field $\eta =\hat{\eta }+\tilde{\eta }=\eta \left(\hat{W}+\tilde{W}\right)$.

The computation of the time derivative then proceeds as follows
(63)$$\begin{array}{c} \frac{d}{dt}V_{\mathrm{neq}}(\tilde{W}∥\hat{W})=-\frac{d}{dt}S_{\hat{\theta }}\left(\tilde{W}∥\hat{W}\right)+\frac{d}{dt}E_{\mathrm{tot}}(\hat{W}+\tilde{W})-\frac{d}{dt}E_{\mathrm{tot}}(\hat{W})-\frac{d}{dt}\int_{\Omega }\rho c_{V}\tilde{\theta }\;dv\\ =-\frac{d}{dt}\int_{\Omega }\rho \hat{\theta }\tilde{\eta }\;dv+\frac{d}{dt}\int_{\Omega }\rho c_{V}\tilde{\theta }\;dv+\frac{d}{dt}E_{\mathrm{tot}}(\hat{W}+\tilde{W})-\frac{d}{dt}E_{\mathrm{tot}}(\hat{W})-\frac{d}{dt}\int_{\Omega }\rho c_{V}\tilde{\theta }\;dv\\ =-\frac{d}{dt}\int_{\Omega }\rho \hat{\theta }\tilde{\eta }\;dv+\frac{d}{dt}E_{\mathrm{tot}}(\hat{W}+\tilde{W})-\frac{d}{dt}E_{\mathrm{tot}}(\hat{W}). \end{array}$$
The time derivatives of the net total energy $E_{\mathrm{tot}}$ can be evaluated using the governing equation for the energy. We get
(64)$$\frac{d}{dt}E_{\mathrm{tot}}(\hat{W}+\tilde{W})=\int_{\Omega }\mathrm{div}\{\kappa \nabla (\hat{\theta }+\tilde{\theta })\}\;dv,\;\frac{d}{dt}E_{\mathrm{tot}}(\hat{W})=\int_{\Omega }\mathrm{div}\left\{\kappa \nabla \hat{\theta }\right\}\;dv,$$
which is a straightforward consequence of ([Disp-formula FD19b-entropy-21-00704]) and the integration over the domain Ω. Using (Equation 64) in (Equation 63) yields
(65)$$\frac{d}{dt}V_{\mathrm{neq}}(\tilde{W}∥\hat{W})=-\frac{d}{dt}\int_{\Omega }\rho \hat{\theta }\tilde{\eta }\;dv+\int_{\Omega }\mathrm{div}(\kappa \nabla \tilde{\theta })\;dv=-\int_{\Omega }\rho \hat{\theta }\frac{\partial \tilde{\eta }}{\partial t}\;dv+\int_{\Omega }\mathrm{div}(\kappa \nabla \tilde{\theta })\;dv.$$
Let us again recall that $\hat{\theta }$ is the non-equilibrium *steady* state, that is $\frac{\partial \hat{\theta }}{\partial t}=0$. Here we explicitly see that the time derivative contains the time derivative of the relative entropy $\tilde{\eta }$ and a flux term. The presence of the time derivative of the relative entropy indicates that the formula for the time derivative will depend on the *entropy production*.

Next, we use the evolution equation for $\tilde{\eta }$, see (Equation 61), and we substitute into the first term in (Equation 65). We get
(66)$$\begin{array}{c} \frac{d}{dt}V_{\mathrm{neq}}(\tilde{W}∥\hat{W})=-\int_{\Omega }\frac{\kappa }{c_{V}^{2}}\hat{\theta }\nabla \tilde{\eta }•\nabla \tilde{\eta }\;dv-\int_{\Omega }\frac{2\kappa }{c_{V}^{2}}\hat{\theta }\nabla \tilde{\eta }•\nabla \hat{\eta }\;dv-\int_{\Omega }\hat{\theta }\mathrm{div}(\frac{\kappa }{c_{V}}\nabla \tilde{\eta })\;dv\\ +\int_{\Omega }\mathrm{div}(\kappa \nabla \tilde{\theta })\;dv. \end{array}$$
Apparently, the first term in (Equation 66) is in the leading order quadratic in the perturbation and it is non-positive. The aim is to manipulate the remaining terms in such a way that the complete right-hand side of (Equation 66) is also in the leading order quadratic in the perturbation. This must be possible, since the functional $V_{\mathrm{neq}}(\tilde{W}∥\hat{W})$ that is being differentiated is in the leading order quadratic in the perturbation, and the governing equation for the perturbation does not contain a zeroth order term.

In our case, we can in fact show that the last three terms on the right-hand side of (Equation 66) vanish, and that the only term that remains on the right-hand side of (Equation 66) is negative for all nonzero perturbations, hence, we get an unconditional stability result. (Dealing with a counterpart of (Equation 66) in a more complex setting than the heat conduction problem one can of course expect the presence of terms that do not have a definite sign. Consequently, the right-hand side of (Equation 66) will be non-positive only if the additional terms can be bounded by the non-positive terms. This will in general lead to conditional stability results.) Let us start manipulating the terms. Recalling that
(67)$$\tilde{\eta }=c_{V}\ln\left(1+\frac{\tilde{\theta }}{\hat{\theta }}\right),\;\hat{\eta }=c_{V}\ln\frac{\hat{\theta }}{\theta_{\mathrm{ref}}},$$
we note that $\tilde{\eta }=0$ whenever $\tilde{\theta }=0$, hence, the boundary condition ([Disp-formula FD38b-entropy-21-00704]) for $\tilde{\theta }$ implies that the relative entropy $\tilde{\eta }$ vanishes on the boundary, ${\left.\tilde{\eta }\right|}_{\partial \Omega }=0$. Further, we see that
(68)$$\nabla \tilde{\eta }=c_{V}\frac{\nabla (\hat{\theta }+\tilde{\theta })}{\hat{\theta }+\tilde{\theta }}-c_{V}\frac{\nabla \hat{\theta }}{\hat{\theta }},\;\nabla \hat{\eta }=c_{V}\frac{\nabla \hat{\theta }}{\hat{\theta }},$$
and we get the following identities
(69a)$$\begin{array}{c} -\hat{\theta }\mathrm{div}\left(\frac{\kappa }{c_{V}}\nabla \tilde{\eta }\right)=\frac{\kappa }{c_{V}}\nabla \tilde{\eta }•\nabla \hat{\theta }-\mathrm{div}\left(\kappa \hat{\theta }\frac{\nabla (\hat{\theta }+\tilde{\theta })}{\hat{\theta }+\tilde{\theta }}-\kappa \nabla \hat{\theta }\right), \end{array}$$
(69b)$$\begin{array}{c} \frac{2\kappa }{c_{V}^{2}}\hat{\theta }\nabla \tilde{\eta }•\nabla \hat{\eta }=\frac{2\kappa }{c_{V}}\nabla \tilde{\eta }•\nabla \hat{\theta }, \end{array}$$
(69c)$$\begin{array}{c} \mathrm{div}(\kappa \nabla \tilde{\theta })=\mathrm{div}\left(\kappa \frac{\hat{\theta }}{\hat{\theta }+\tilde{\theta }}\nabla \tilde{\theta }+\kappa \frac{\tilde{\theta }}{\hat{\theta }+\tilde{\theta }}\nabla \tilde{\theta }\right), \end{array}$$
(69d)$$\begin{array}{c} \mathrm{div}\left(\kappa \frac{\hat{\theta }}{\hat{\theta }+\tilde{\theta }}\nabla \hat{\theta }\right)=\mathrm{div}(\kappa \nabla \hat{\theta })-\mathrm{div}\left(\kappa \frac{\tilde{\theta }}{\hat{\theta }+\tilde{\theta }}\nabla \hat{\theta }\right). \end{array}$$
Note that the identities simplify considerably if we use the fact that $\hat{\theta }$ is a solution to $\mathrm{div}(\kappa \nabla \hat{\theta })=0$. Using identites (69), we see that
(70)$$\begin{array}{cc} -\int_{\Omega }\frac{2\kappa }{c_{V}^{2}}\hat{\theta }\nabla \tilde{\eta }•\nabla \hat{\eta }\;dv-\int_{\Omega }\hat{\theta }\mathrm{div}\left(\frac{\kappa }{c_{V}}\nabla \tilde{\eta }\right)\;dv & +\int_{\Omega }\mathrm{div}(\kappa \nabla \tilde{\theta })\;dv\\ & =\int_{\Omega }\mathrm{div}\left(\kappa \tilde{\theta }\frac{\nabla (\hat{\theta }+\tilde{\theta })}{\hat{\theta }+\tilde{\theta }}\right)\;dv-\int_{\Omega }\frac{\kappa }{c_{V}}\nabla \tilde{\eta }•\nabla \hat{\theta }\;dv=0, \end{array}$$
where the first integral vanishes in the virtue of the Stokes theorem and the boundary condition ${\left.\tilde{\theta }\right|}_{\partial \Omega }=0$, while the second integral vanishes due to the integration *by parts*, where we exploit the boundary condition ${\left.\tilde{\eta }\right|}_{\partial \Omega }=0$ and the fact that $\hat{\theta }$ is a solution to $\mathrm{div}(\kappa \nabla \hat{\theta })=0$.

Using (Equation 70) we can conclude that the formula for the time derivative (Equation 66) simplifies to
(71)$$\frac{d}{dt}V_{\mathrm{neq}}(\tilde{W}∥\hat{W})=-\int_{\Omega }\frac{\kappa }{c_{V}^{2}}\hat{\theta }\nabla \tilde{\eta }•\nabla \tilde{\eta }\;dv.$$
The time derivative of $V_{\mathrm{neq}}(\tilde{W}∥\hat{W})$ is negative unless $\tilde{\eta }$ is equal to zero everywhere in Ω. This means that $V_{\mathrm{neq}}(\tilde{W}∥\hat{W})$ is indeed a Lyapunov type functional suitable for the nonlinear stability analysis of the steady non-equilibrium temperature field $\hat{\theta }$. Consequently, we see that the steady state $\hat{\theta }$ is unconditionally asymptotically stable.

#### 5.3.7. Relation to the Standard Energy Method

We can again note that if the temperature perturbation $\tilde{\theta }$ is small in the sense that $\frac{\tilde{\theta }}{\hat{\theta }}<<1$, then
(72)$$V_{\mathrm{neq}}(\tilde{W}∥\hat{W})=\int_{\Omega }\rho c_{V}\hat{\theta }\left[\frac{\tilde{\theta }}{\hat{\theta }}-\ln\left(1+\frac{\tilde{\theta }}{\hat{\theta }}\right)\right]\;dv\approx \int_{\Omega }\rho c_{V}\frac{{\tilde{\theta }}^{2}}{\hat{\theta }}\;dv,$$
hence, $V_{\mathrm{neq}}$ is almost equal to the (square of) the weighted $L^{2}\left(\Omega \right)$ norm of the perturbation temperature field $\tilde{\theta }$. Moreover, if $\hat{\theta }$ is position independent, that is if we analyse the *equilibrium* rest state, we recover, up to a constant, the standard *energy method* functional $V_{\mathrm{std}}$, see (Equation 12).

Further, we see that the integrand in the standard Lyapunov functional $V_{\mathrm{std}}$ is insensitive to the direction of the deviation from the non-equilibrium rest state. The integrand takes the same value both for $\tilde{\theta }$ and $-\tilde{\theta }$. On the other hand, the integrand in the Lyapunov type functional (Equation 56) does not have this property. Its value is different for $-\tilde{\theta }$ and $\tilde{\theta }$, and, moreover, its value approaches infinity as $\tilde{\theta }\to -\hat{\theta }$.

One can also note that the *relative entropy* functional, that is the functional $S_{\mathrm{rel}}{=}_{\mathrm{def}}\int_{\Omega }\rho \tilde{\eta }\;dv$, see (Equation 59) and (Equation 60), can not be used as a Lyapunov type functional. It does not provide a control on the spatial distribution of the perturbations. In particular, $S_{\mathrm{rel}}=0$ does not imply that $\tilde{\theta }=0$ in the whole domain Ω.

#### 5.3.8. Weak—Strong Uniqueness Property

The notion of stability can be also understood as “continuous dependence of thermodynamical processes upon initial state and supply terms”, see Dafermos [46], which is a different concept than that we have discussed above. In particular, the aim of the stability analysis understood in this sense is to show that if two solutions to a given initial-boundary value problem *share the same initial condition and boundary condition*, then they coincide also for later times. This is a nontrivial question when the two solutions sharing the same initial conditions are for example the strong and the weak solution. Interestingly, a similar construction of a “distance” functional as outlined above have been used on *ad hoc* grounds byFeireisl et al. [47], Feireisl and Novotný [48] in their seminal analysis of weak–strong uniqueness property for the Navier–Stokes–Fourier system. (See also Feireisl and Novotný [49].) Note however that the weak–strong uniqueness analysis by Feireisl et al. [47], Feireisl and Novotný [48] is again restricted to a *thermodynamically isolated* system, and it has no implications for the nonlinear stability analysis in the sense we are using in the present contribution.

## 6. Conclusions

We have shown that a Lyapunov type functional suitable for the nonlinear stability analysis of steady solutions to the heat conduction problem in a rigid body can be constructed with the use of thermodynamical concepts. In particular, the thermodynamical concepts have been shown to be useful even in the case of a nonlinear stability analysis of a *thermodynamically open* system.

The outlined construction of the physically motivated Lyapunov type functional is rather superfluous given *the simple setting we have been studying*. In the present case, the standard *energy method* definitely provides a formally much simpler approach to the nonlinear stability analysis. The construction however shows that nonlinear stability analysis can be indeed based on an insight into the physics behind the given system of governing equations. In particular, it indicates that using the square of the Lebesgue space norm of the temperature field as a Lyapunov functional is just a matter of mathematical convenience. The physically motivated Lyapunov type functional is different from the mathematically convenient one.

More importantly, the outlined construction of a physically well-motivated Lyapunov type functional seems to be *general enough to be applied even in a more complex thermomechanical settings*. (Here, however, one can not in general expect unconditional stability, the non-equilibrium steady state must be expected to be stable only for some parameter values/size of the initial perturbation and so forth.) Indeed, the construction of the Lyapunov type functional is in fact based only on the knowledge of the specific Helmholtz free energy ψ, which is known for many complex materials such as polymeric fluids. In such complicated settings the apparent complexity of the outlined construction could turn into an advantage, since the Lebesgue norm ${∥\cdot ∥}_{L^{2}\left(\Omega \right)}$ used in the standard *energy method* does not respect the natural physical background of the corresponding governing equations. Consequently, the advocated approach could provide a tool for nonlinear stability analysis in complex thermomechanical systems for which the standard *energy method* has been so far unsuccessful.

The reader interested in more involved applications of the proposed construction is kindly referred to Appendix A, where we discuss the stability the spatially inhomogeneous steady temperature field in a rigid body wherein the heat conductivity is a function of the temperature. Application to the flows of polymeric fluids is discussed in Dostalík et al. [50], where the authors analyse the stability of the Taylor–Couette type flow of the Giesekus fluid. Finally, Dostalík and Průša [51] use the proposed method in a coupled thermal convection/conduction setting.

## Figures and Tables

**Figure 1 entropy-21-00704-f001:**
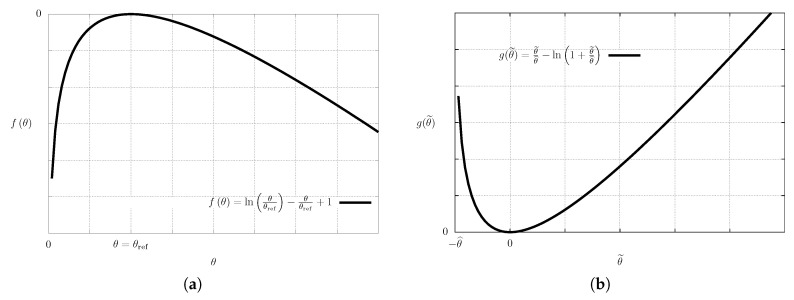
Auxiliary functions. (**a**) Plot of function f(θ) that appears as the integrand in (Equation 31) and (Equation 34); (**b**) Plot of function $g(\tilde{\theta })$ that appears as the integrand in (Equation 54).

**Figure 2 entropy-21-00704-f002:**
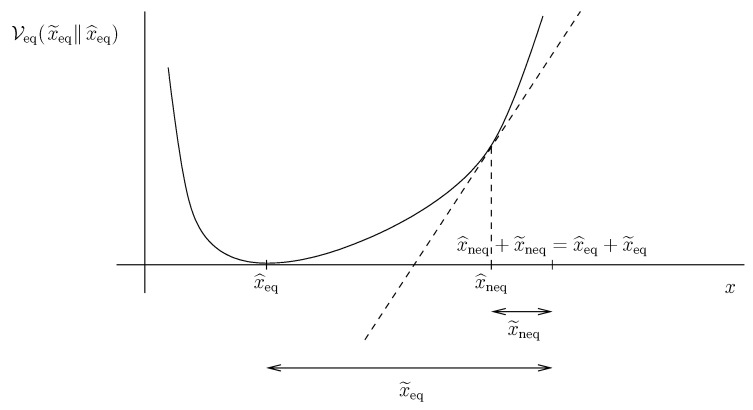
Construction of the Lyapunov functional $V_{\mathrm{neq}}\left({\tilde{x}}_{\mathrm{neq}}∥{\hat{x}}_{\mathrm{neq}}\right)$ for a non-equilibrium state ${\hat{x}}_{\mathrm{neq}}$ from the Lyapunov functional $V_{\mathrm{eq}}\left({\tilde{x}}_{\mathrm{eq}}∥{\hat{x}}_{\mathrm{eq}}\right)$ for the rest state ${\hat{x}}_{\mathrm{eq}}$.

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
