# Peer review of "Thermodynamics and Stability of Non-Equilibrium Steady States in Open Systems"

_entropy, 2019, doi:10.3390/e21070704_

Round 1

Reviewer 1 Report

The paper is focused on the misuse of the word "energy" in the literature pertaining to the stability of non-equilibrium steady states. A more appropriate (from the physical point of view) Lyapunov functional for the stability analysis of these states is introduced. A simple example to show the connections between the results predicted by employing the novel Lyapunov functional and that of the "energy" method is provided. The paper is well written and thus I recommend publication in Entropy. I only suggest the Authors to indicate the physical dimensions of the specific Helholtz free energy "psi" and not the unit of measurement just before equation (13). In other words [psi]=J/kg should be replaced with [psi]=L T^(-2). 

Author Response

We thank the reviewer for reading the manuscript and the comments.

POINT 1

I only suggest the Authors to indicate the physical dimensions of the specific Helholtz free energy "psi" and not the unit of measurement just before equation (13). In other words [psi]=J/kg should be replaced with [psi]=L T^(-2).

RESPONSE 1

We have rewritten all statements concerning the units in the form standard in dimensional analysis. The unit of the specific Helmholtz free energy [psi] is now written as [psi] = L^2/T^2 as requested. The same applies to the specific entropy, internal energy and so forth at various locations in the manuscript.

Reviewer 2 Report

I reject the paper because I doubt that it or similar work has already been published.

The paper has  already been cited by:

Málek, J., Průša, V., Skřivan, T., & Süli, E. (2018). Thermodynamics of viscoelastic rate-type fluids with stress diffusion. Physics of Fluids30(2), 023101.

https://aip.scitation.org/doi/abs/10.1063/1.5018172

Author Response

POINT 1

I reject the paper because I doubt that it or similar work has already been published.

The paper has  already been cited by:

Málek, J., Průša, V., Skřivan, T., & Süli, E. (2018). Thermodynamics of viscoelastic rate-type fluids with stress diffusion. Physics of Fluids30(2), 023101.

https://aip.scitation.org/doi/abs/10.1063/1.5018172

RESPONSE 1

A very early version of the manuscript has been submitted to arXiv (https://arxiv.org/abs/1709.05968). The manuscript or any part thereof has not been published in any scholarly journal.

The paper "Málek, J., Průša, V., Skřivan, T., & Süli, E. (2018). Thermodynamics of viscoelastic rate-type fluids with stress diffusion. Physics of Fluids30(2), 023101." has been written by some authors of the current manuscript. The very early arXiv version of the manuscript is in the paper cited as follows: "See also Bulı́ček et al. (2017) for a proposal concerning the possible extension of this procedure to stability analysis of non-equilibrium steady states in thermodynamically open systems." No part of the current manuscript has been used in the paper "Málek, J., Průša, V., Skřivan, T., & Süli, E. (2018). Thermodynamics of viscoelastic rate-type fluids with stress diffusion. Physics of Fluids30(2), 023101."

Round 2

Reviewer 2 Report

The reviewer thinks that the paper is interesting, but some explanations and mathematical proofs are explained too detailed. For example, there are almost 100 equations. This type of writing style is adequate to books. In the reviewer opinion's, If the authors could synthesize the mathematical proofs, the paper will be more readable. 

Another point is the appendix. The number of pages is too higher.

 I think that the editor and authors could consider dividing the paper into two articles — part I about the theoretical and part II with the applications (Appendix).

The symbol eta for the specific entropy is not usual. A nomenclature could be useful.

Author Response

We thank the reviewer for the comments. The response follows below.

POINT 1

The reviewer thinks that the paper is interesting, but some explanations and mathematical proofs are explained too detailed. For example, there are almost 100 equations. This type of writing style is adequate to books. In the reviewer opinion's, If the authors could synthesize the mathematical proofs, the paper will be more readable.

RESPONSE 1

We agree that the the explanations are detailed, which in fact was our objective. We wanted the necessary algebraic manipulations to be described in detail and easy to follow. We wanted to make the manuscript more accessible to the general audience that may not be used to this kind of manipulations. Perhaps the writing style is a matter of opinion.

Note also that some of the equations are in fact definitions, hence the number of equations is quite high.

POINT 2

Another point is the appendix. The number of pages is too higher.

I think that the editor and authors could consider dividing the paper into two articles — part I about the theoretical and part II with the applications (Appendix).

RESPONSE 2

We believe it is important to keep the manuscript as it is. Virtually any technique in the stability analysis will work in the simple case of linear parabolic equation that is discussed in the main text body. Consequently, it is vital to show that the proposed approach works seamlessly also in the nonlinear setting, which is done in the Appendix. If the main text body and the appendix are separated, the message that the proposed approach is especially suitable for nonlinear systems would be, in our opinion, considerably weakened.

However, if the editor wanted us to split the manuscript, we would accept it.

POINT 3

The symbol eta for the specific entropy is not usual. A nomenclature could be useful.

RESPONSE 3

The notation eta for the specific entropy is used in the classical treatise Truesdell, C., & Toupin, R. (1960). The classical field theories. In Principles of classical mechanics and field theory/Prinzipien der Klassischen Mechanik und Feldtheorie (pp. 226-858). Springer, Berlin, Heidelberg or in the reference work by Truesdell, C., & Noll, W. (1965). The non-linear field theories of mechanics. In The non-linear field theories of mechanics (pp. 1-579). Springer, Berlin, Heidelberg as well as in modern textbooks such as Gurtin, M. E., Fried, E., & Anand, L. (2010). The mechanics and thermodynamics of continua. Cambridge University Press. This makes the notation standard at least in a large part of continuum mechanics community.

One of the reasons for the high number of equations in the manuscript is that we do not use inline equations. This allows one, in our opinion, to quickly locate where the symbol is used for the first time, and it effectively replaces the nomenclature.